# Extended-spectrum β-lactamase-producing *Enterobacterales* among people living with human immunodeficiency virus across the globe: A systematic review and meta-analysis

Mitkie Tigabie[1]*, Abebe Birhanu[1], Muluneh Assefa[1], Getu Girmay[2], Kebebe Tadesse[3]

1 Department of Medical Microbiology, School of Biomedical and Laboratory Sciences, College of Medicine and Health Sciences, University of Gondar, Gondar, Ethiopia, 2 Department of Immunology and Molecular Biology, School of Biomedical and Laboratory Sciences, College of Medicine and Health Sciences, University of Gondar, Ethiopia, 3 Department of Medical Laboratory Sciences, Pawe Health Science College, Pawe, Ethiopia

* mitku1621@gmail.com

## Abstract

### Background

People living with HIV are vulnerable to antibiotic-resistant bacterial infections because of frequent healthcare visits, the consumption of many antimicrobials, and the weakened immune system to fight infections. Our objective was to provide comprehensive data about ESBL-producing *Enterobacterales* among HIV-positive individuals across the globe.

### Methods

This meta-analysis was conducted in accordance with the Preferred Reporting Items for Systematic Reviews and Meta-Analyses (PRISMA) guidelines. To select eligible articles published between January 1, 2010, and May 12, 2024, a literature search was performed on available electronic databases such as PubMed, Hinari, Google Scholar, and Scopus. The quality of the included studies was assessed via the Joanna Briggs Institute critical appraisal tool. The data were extracted from the eligible studies via Microsoft Excel 2019 and analyzed via STATA version 17. A random effects model was constructed via the DerSimonian and Laird method. The heterogeneity was checked through the Cochrane Q statistic, and the magnitude was quantitatively measured via $I^2$ statistics. To determine the possible sources of heterogeneity, a subgroup analysis was performed. Additionally, a sensitivity analysis was conducted, and publication bias was checked via funnel plots and Egger's regression tests. A p value of less than 0.05 was considered evidence of heterogeneity and small study effects according to the Cochrane Q statistic and Egger's test, respectively. The protocol was registered previously (PROSPERO ID: CRD42024557981).

**Data availability statement:** All relevant data are within the manuscript and its Supporting Information files.

**Funding:** The author(s) received no specific funding for this work.

**Competing interests:** The authors have declared that no competing interests exist.

**List of abbreviations** AMR Antimicrobial resistance ART Antiretroviral therapy CDT Combination disk test DDST Double-disk synergy test ESBL Extended-spectrum β-lactamase HIV Human immunodeficiency virus MDR Multidrug resistance WHO World Health Organization.

## Results

A total of 5305 HIV-positive individuals from 20 studies were included in our meta-analysis. The overall pooled prevalence of ESBL-producing *Enterobacterales* among HIV-positive individuals was 20.30% (931/5305; 95% CI: 15.13–25.47%, P < 0.001), with a high level of heterogeneity ($I^2 = 97.82\%$, P < 0.001). The predominant ESBL producers were *K. pneumoniae*, with a pooled prevalence of 40.84% (76/217; 95% CI: 26.87–54.81%), followed closely by *E. coli* at 40.14% (348/985; 95% CI: 27.83–52.45%). In the subgroup analysis, the highest magnitude of ESBL-producing pathogens was observed in Asia (195/715; 28.55%), followed by Africa (666/3981; 19.12%). Additionally, the highest pooled prevalence of ESBL-producing pathogens among HIV-positive individuals was reported to be colonization 23.78% (613/2455; 95% CI: 15.36–32.19, $I^2 = 96.78\%$, p < 0.001), followed by infection 15.77% (318/2850; 95% CI: 10.06–21.49, $I^2 = 97.45\%$, p < 0.001). Among the different types of ESBL enzyme-encoding genes, $bla_{CTX-M}$ was the most common (73 out of 150 isolates, 48.7%), followed by $bla_{TEM}$ (49 out of 150, 32.7%).

## Conclusion and recommendations

This study demonstrated that HIV-positive individuals are commonly colonized and infected with ESBL-producing *Enterobacterales*. The highest prevalence of these pathogens was reported in Asia and Africa. To reduce mortality from severe bacterial infections in HIV patients, resources should be distributed equitably across all regions. Particular attention should be given to high-prevalence areas, where early detection of colonization and infection with antibiotic-resistant pathogens is critical. Enhanced surveillance of ESBL-producing organisms among HIV-positive individuals is also strongly recommended.

## Introduction

Antimicrobial resistance (AMR) is one of the top ten threats to global public health, and it is a major and serious issue in the 21st century [1]. Recent studies have provided updated insights into the global impact of antimicrobial resistance (AMR). In 2021, approximately 1.14 million deaths were directly attributable to bacterial AMR, with a total of 4.71 million deaths associated with AMR worldwide [2]. Projections indicate that, without significant intervention, AMR could be responsible for up to 10 million deaths annually by 2050 [3]. Regionally, the World Health Organization (WHO) African Region experiences a significant impact, with approximately 250,000 deaths attributable to bacterial AMR infections [4]. In the European region, about 133,000 deaths are attributed to bacterial AMR infections [5].

Although the overuse and misuse of antimicrobials in humans, animals and their environments are the main factors that increase AMR, limited AMR surveillance and poor infection prevention and control practices also contribute to the increase in AMR infections [6]. Furthermore, immunosuppressive conditions such as human immunodeficiency virus (HIV) infection create favorable conditions for the development

of bacterial resistance. People living with HIV are more likely to have contact with healthcare facilities and be exposed to invasive medical procedures than the general population is. Moreover, these populations are vulnerable to bacterial AMR infections due to frequent hospital admissions, the consumption of many antimicrobial agents, and the weakened immune system to fight infections [7,8]. Additionally, antibiotic prophylaxis for the prevention of opportunistic infections among HIV patients is the major factor contributing to the development of resistance [9].

Human immunodeficiency virus remains a major global public health concern, with a significant burden in many low- and middle-income countries. As of 2023, an estimated 39.9 million people were living with HIV worldwide, and approximately 630,000 deaths were attributed to HIV-related causes. The cumulative number of individuals living with HIV has continued to rise, reaching an estimated 42.3 million to date. This persistent and widespread prevalence of HIV not only presents challenges for disease management and health systems but also has important implications for other infectious diseases, particularly in relation to antimicrobial resistance [10].

Acquired immunodeficiency syndrome (AIDS) occurs at the most advanced stage of HIV infection, and in this stage, people can also develop not only opportunistic infections such as tuberculosis, pneumocystis pneumonia and cryptococcal meningitis but also severe bacterial infections [11,12]. These severe bacterial infections are caused by common bacterial pathogens that are grouped under gram-positive cocci and gram-negative bacilli [12]. Among bacterial pathogens, *Enterobacterales* are multidrug resistant owing to their ability to produce various resistance mechanisms, including efflux pumps, porin modification, overexpression, and enzyme production [13]. Enzymatic inactivation due to extended-spectrum β-lactamase (ESBL) enzyme production is the predominant cause of resistance to β-lactam antibiotics [13,14]. These enzymes can hydrolyze penicillins, cephalosporins, and aztreonam. Penicillins and cephalosporins are widely accessible and frequently used to treat various infections globally, especially in developing countries. [14].

*Escherichia coli* (*E. coli*) and *Klebsiella pneumoniae* (*K. pneumoniae*) are the predominant *Enterobacterales* that produce ESBL enzymes [14,15]. Infections with these pathogens increase the risk of treatment failure and have led to increased use of last-resort antibiotics such as carbapenems [16] and combination therapies with more toxic antibiotics such as polymyxin, which cause nephrotoxicity, ototoxicity, and neuromuscular blockade [17].

To improve the management of HIV-positive patients and reduce mortality due to complications with bacterial infections, comprehensive data concerning resistant bacteria due to ESBL enzyme inactivation of antibiotics are paramount. Nevertheless, on the basis of previous studies, the proportion of ESBL-producing *Enterobacterales* among HIV-positive individuals varies from 2.3% to 57.4% [18,19]. In addition, there is a great discrepancy in the prevalence of ESBL-producing *Enterobacterales* among HIV-positive individuals in previously published studies [19–31].

Moreover, most existing systematic reviews and meta-analyses on HIV-related infections have focused on multidrug-resistant *Mycobacterium tuberculosis* [32] and methicillin-resistant *Staphylococcus aureus* [33]. However, pooled data on ESBL-producing *Enterobacterales* among HIV-positive individuals are still lacking. Therefore, this systematic review and meta-analysis aimed to provide comprehensive data concerning ESBL-producing *Enterobacterales* among HIV-positive individuals across the globe.

## Methods

### Study design and protocol registration

This systematic review and meta-analysis was performed in accordance with the Preferred Reporting Items for Systematic Reviews and Meta-Analyses (PRISMA) guidelines (**SF 1 Table**). The protocol is available on the International Prospective Register of Systematic Reviews (PROSPERO ID: CRD42024557981).

### Search strategy

Two authors (MT, KT) searched the Medline/PubMed, Hinari, Google Scholar, and Scopus electronic databases for studies published from 1 January 2010–12 May 2024. We used keywords alone and in combination with Boolean

operators such as "OR" or "AND". For example, articles were identified via MeSH terms from keywords of the title on Medline/PubMed, as follows (((((((antimicrobial resistance) OR (antibiotic resistance)) OR (multidrug resistance)) OR (extended-spectrum β-lactamase)) OR (ESBL)) AND (((((((gram-negative) OR (bacteria)) OR (bacilli)) OR (enterobacteriaceae)) OR (enterobacterales)) OR (*Escherichia coli*)) OR (*Klebsiella pneumoniae*))) AND (((((immunocompromised host) OR (human immunodeficiency virus)) OR (acquired immunodeficiency syndrome)) OR (HIV)) OR (HIV/AIDS))) AND ("2010/01/01" to "2024/05/12").

### Outcome of interest

The main outcome was the prevalence of ESBL-producing *Enterobacterales* among people living with HIV across the globe. We estimated the prevalence by dividing the number of ESBL-producing *Enterobacterales* cases by the total sample size. For case-control studies, the prevalence was calculated only among the cases. The prevalence of ESBL-producing bacteria for individual species was calculated by dividing the number of ESBL-producing isolates by the total number of isolates for that specific species.

### Eligibility criteria

We used the CoCoPop (Condition, Context, and Population) approach, in which the prevalence of ESBL-producing *Enterobacterales* was considered the condition (CO), people living with HIV were considered the population (POP), and the world served as the context (CO). To identify eligible articles, we declared the predetermined inclusion and exclusion criteria; all cross-sectional, case–control, and cohort studies reported ESBL-producing *Enterobacterales* among people living with HIV worldwide. The review included studies with mixed populations which reported on ESBL-producing *Enterobacterales* prevalence by HIV status. Studies published between January 1, 2010, and May 12, 2024, which were written exclusively in the English language, and studies that were peer reviewed were included in this systematic review and meta-analysis. However, systematic reviews and meta-analyses, case reports, case series, conference papers and pilot studies were excluded. Preprint studies were also excluded.

### Quality assessment

Three reviewers (MT, KT, and AB) independently and in duplicate screened the titles and abstracts of the studies and subsequently assessed the potential eligibility of the relevant full texts on the basis of the predefined inclusion criteria. The quality of studies was assessed via standard critical appraisal tools prepared by the Joanna Briggs Institute (JBI) for prevalence and case–control studies [34]. The JBI appraisal checklist contains 9 and 10 questions for cross-sectional and case–control studies, respectively. These critical appraisal tools have yes, 'no', 'unclear', and 'not applicable' options. For each question, a score of 0 was assigned for 'no', 'unclear', and 'not applicable' and a score of 1 was assigned for 'yes'. The discrepancies were solved by taking the average score. The total score is calculated by counting the number of "yes" in each row. On the basis of the score of the quality assessment tool, the highest score had the minimum risk of bias. Overall scores ranging from 0–4, 5–6, and 7–9 for prevalence studies and from 0–4, 5–7, and 8–10 for case–control studies are declared high, moderate, and low risk of bias, respectively [35]. Finally, studies with a score of five and above for "yes" (have moderate and low risk of bias) were included in the systematic review and meta-analysis (**SF 2 Table**).

### Data extraction

All of the studies obtained from different electronic databases were combined and properly exported to EndNote version 9.2 (Clarivate Analytics, Philadelphia, PA, USA). Then, the articles were merged into one folder for identification, duplicate articles were removed, and the quality of the studies was checked. We subsequently assessed the eligibility of the studies imported into Microsoft Excel 2019 (Microsoft Corp., Redmond, WA, USA).

All important parameters were extracted from each study by three authors (MT, KT, & AB.) independently. Discrepancies between them were resolved by consensus. The data extraction format was prepared according to the Preferred Reporting Items for Systematic Reviews and Meta Analyses (PRISMA) guidelines. For each study, the primary author, year of publication, sample size, study design, study year, study country, age group, investigation type (phenotype or genotype), status (infection or colonization) and methods of detection for ESBL were extracted. Furthermore, data on the total number of ESBL-producing isolates, individual ESBL-producing species, and their corresponding total number of isolates were extracted.

## Statistical analysis

The extracted data were exported to STATA software version 17 for analysis. We conducted a meta-analysis via the random effects DerSimonian and Laird model to estimate the pooled prevalence and 95% confidence intervals (CIs) [36]. The presence of between-study heterogeneity was checked by using the Cochrane Q statistic. The magnitude of heterogeneity between the included studies was quantitatively measured by the inverse variance ($I^2$ statistic). $I^2 = 0$, $I^2 = 0$–25%, $I^2 = 50$–75%, and $I^2 > 75$% indicate no, low, moderate, and high heterogeneity, respectively. The significance of heterogeneity was determined by the p value of the Cochrane Q statistic, and a p value of less than 0.05 was evidence of heterogeneity [37]. The possible sources of heterogeneity were further investigated by performing a subgroup analysis in reference to the continents, publication year, age categories, and methods of confirmation for the ESBL. Additionally, a sensitivity analysis was conducted to determine the influence of single studies on the pooled estimates. Publication bias was checked by using a funnel plot test graphically and more objectively through Egger's regression tests. A statistically significant Egger's test (P value < 0.05) indicates the presence of a small study effect [38].

## Results

### Study selection and identification

We identified a total of 1478 articles from available scientific databases such as PubMed, Hinari, Google Scholar, and Scopus. Of these, 997 studies were removed because they were duplications. Four hundred eighty-one studies were screened by reading their titles and abstracts. Two hundred forty-six articles were removed due to unrelated topics, incorrect publication periods, mixed or non-human populations, or inclusion of non-*Enterobacterales* or other pathogens. After that, 235 articles were screened by reading their full texts, and 198 studies were excluded because they were pilot studies, preprints, or involved study participants other than HIV patients. Thirty-seven studies passed the eligibility assessment, but 17 articles were removed because the outcomes were not reported or did not fulfill the quality criteria. Finally, 20 studies were eligible and included in the final meta-analysis [18–21,23–31,39–45], as presented in the PRISMA flow diagram (**Fig 1**).

### Summary of the risk of bias results

Out of the thirty-seven studies that passed the eligibility assessment, 11 were excluded because they did not meet the quality criteria. Among the studies that met the quality criteria, 13 out of 20 (65.0%) had a low risk of bias, while 7 (35.0%) had a moderate risk of bias. Six studies were excluded due to a high risk of bias (see **SF 2 Table**).

### Characteristics of the studies included in the systematic review and meta-analysis

Among the included studies, 15 (75.0%) studies were published in and after 2020 [18–21,23–28,39,40,43–45]. Eighteen (90.0%) studies were cross-sectional [18–21,23–31,39,40,43–45], whereas the remaining two studies were case–control studies [30,41]. Eleven (55.0%) studies focused on colonization [20,21,23,24,28–30,39,42,43,45], whereas the remaining 9 studies focused on infection [18,19,25–27,31,40,41,44]. All of the studies included in this review were from three continents: Africa [18,20–23,25,27,29,31,39,40,43–45], Asia [19,26,28,30,41], and Europe [24,42] (**Table 1**).

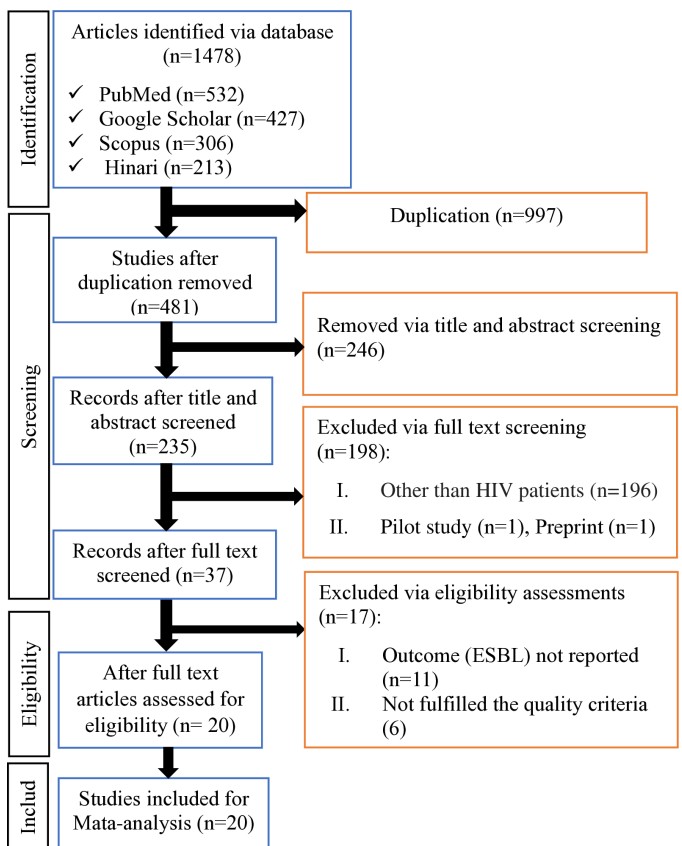

**Fig 1. A flow diagram of study selection for systematic review and meta-analysis of ESBL-producing _Enterobacterales_ among HIV-positive individuals.**

A total of 5305 HIV-positive individuals from 20 studies were included in the meta-analysis of ESBL-producing _Enterobacterales_. Among 20 studies, only one detected ESBL by using Vitek-2 [42], but others used the combination disk test (CDT) or double disk synergy test (DDST). The minimum and maximum sample sizes were 50 [41] and 948 [31], respectively. A minimum (2.33%) and maximum (57.4%) prevalence of ESBL-producing _Enterobacterales_ were reported by Endalamaw et al. [18] and M.R. Rameshkumar et al. [19], respectively (**Table 1**).

## Pooled prevalence of ESBL-producing _Enterobacterales_ among HIV positive individuals

The overall pooled prevalence of ESBL-producing _Enterobacterales_ among HIV-positive individuals was 20.30% (931/5305; 95% CI: 15.13–25.47%, P < 0.001), with a high level of heterogeneity ($I^2 = 97.82\%$, P < 0.001), as presented in (**Fig 2**).

Moreover, the pooled prevalence of ESBL-producing _Enterobacterales_ among HIV-positive individuals with infection was 15.77% (318/2850; 95% CI: 10.06–21.49, $I^2 = 97.45\%$, $p < 0.001$) (**Fig 3**), whereas among asymptomatic HIV-positive individuals, it was 23.78% (613/2455; 95% CI: 15.36–32.19, $I^2 = 96.78\%$, p < 0.001) (**Fig 4**).

The predominant ESBL-producing pathogens among HIV-positive individuals were _K. pneumoniae_, with a pooled prevalence of 40.84% (76/217; 95% CI: 26.87–54.81%) (**Fig 5**), followed closely by _E. coli_ at 40.14% (348/985; 95% CI: 27.83–52.45%) (**Fig 6**), and _Proteus species_ at 37.2% (24/74; 95% CI: 8.03–66.36%) (**SF 4 Fig**). In contrast, _Citrobacter_ species were the least commonly identified ESBL producers, with a prevalence of 9.68% (5/55; 95% CI: 1.26–18.10%) (**SF 4 Fig**). Three studies reported ESBL production in _Salmonella_ spp. The pooled prevalence of ESBL-producing _Salmonella_ spp. was 24.52% (10/42; 95% CI: 7.04–42.00; $I^2 = 94.71\%$, p ≤ 0.019) (**SF 4 Fig**).

**Table 1. Characteristics of individual studies included in the meta-analysis of ESBL-producing *Enterobacterales* among HIV-positive individuals across the globe, 2024.**

| Study (Author, Year) | Study Year | Country | Design | Sample size | Specimen type | Age group | Status | Inv. M* | Meth** | ESBL case N (%) |
|---|---|---|---|---|---|---|---|---|---|---|
| Bayleyegn et al, 2021 [39] | 2020 | Ethiopia | CS | 161 | Stool | Children | Colonization | P | CDT | 32 (19.9) |
| Dimani et al, 2023 [20] | 2021 | Cameroon | CS | 185 | Stool and rectal swabs | All age | Colonization | P&G | DDST | 61 (32.9) |
| Endalamaw et al, 2020 [18] | 2017 | Ethiopia | CS | 387 | Urine | All age | Infection | P | DDST | 9 (2.3) |
| Falodun et al, 2021 [21] | 2017 | Nigeria | CS | 100 | Stool | Adult | Colonization | P | DDST | 56 (56.0) |
| Jerry et al, 2021 [40] | 2017 | Nigeria | CS | 205 | Urine | All age | Infection | P | DDST | 23 (11.2) |
| John-Onwe et al, 2022 [25] | 2019 | Nigeria | CS | 200 | Urine | All age | Infection | P | DDST | 58 (29.0) |
| M.R. Rameshkumar et al, 2021 [19] | NR | India | CS | 183 | Urine, pus, sputum, blood, and vaginal swabs | All age | Infection | P&G | CDT | 105 (57.4) |
| Maharjan et al, 2022 [26] | 2019 | Nepal | CS | 263 | Sputum | All age | Infection | P&G | CDT | 19 (7.2) |
| Manyahi et al, 2020 [23] | 2017-2018 | Tanzania | CS | 595 | Rectal swabs | Adult | Colonization | P&G | DDST | 244 (41.0) |
| Nwokolo et al, 2022 [27] | NR | Nigeria | CS | 363 | Urine | All age | Infection | P&G | CDT | 44 (12.1) |
| Osazuwa et al, 2011 [31] | 2009-2010 | Nigeria | CS | 948 | Urine & diarrheal stools | All age | Infection | P | DDST | 38 (4.0) |
| Padmavathy et al, 2011 [41] | NR | India | CC | 50 | Urine | All age | Infection | P | DDST | 13 (26.0) |
| Reinheimer et al, 2017 [42] | 2014-2016 | Germany | CC | 109 | Rectal swabs | Adult | Colonization | P | Vitek-2 | 7 (6.4) |
| Said et al, 2022 [43] | 2021 | Tanzania | CS | 236 | Stool and rectal swabs | Children | Colonization | P&G | CDT | 56 (23.7) |
| Simeneh et al, 2022 [44] | 2021 | Ethiopia | CS | 251 | Urine | Adult | Infection | P | DDST | 9 (3.6) |
| Singh et al, 2020 [28] | 2017-2018 | India | CS | 100 | Oral swabs | Adult | Colonization | P&G | CDT | 14 (14.0) |
| Subramanya et al, 2019 [30] | 2016-2017 | Nepal | CS | 119 | Rectal swabs | All age | Colonization | P&G | DDST | 46 (38.7) |
| Surgers et al, 2022 [24] | 2018-2019 | France | CS | 500 | Rectal swabs | Adult | Colonization | P&G | CDT | 61 (12.2) |
| Wilmore et al, 2017 [29] | 2014-2015 | Zimbabwe | CS | 175 | Stool | Children | Colonization | P | CDT | 24 (13.7) |
| 12 et al, 2022 [45] | 2021 | Cameroon | CS | 175 | Vaginal swabs | Adult | Colonization | P&G | DDST | 12 (6.9) |

Note: Inv. M

* = Investigation methods, Meth

** = Method used for phenotypic ESBL confirmation, CS = Cross-sectional, CC = Case–control, CDT = Combination disk test, DDST = Double–disc synergy test, NR = Not reported, P = Phenotypic, G = Genotypic, ESBL = Extended-spectrum β-lactamase.

Regarding infection and colonization with ESBL-producing *E. coli* and *K. pneumoniae* among HIV-positive individuals, comparable pooled prevalence rates were observed in both infection and colonization cases. For infection cases, the prevalence of ESBL-producing *E. coli* was 37.0% (237/537; 95% CI: 20.08–53.94%), while that of *K. pneumoniae* was 38.76% (42/119; 95% CI: 22.69–54.83%) (**SF 4 Fig**). Similarly, among colonization cases, the prevalence of ESBL-producing *E. coli* was 42.93% (343/448; 95% CI: 24.16–61.69%), and that of *K. pneumoniae* was 42.36% (34/98; 95% CI: 15.97–68.74%) (**SF 4 Fig**).

## Publication bias

Publication bias was assessed to determine bias related to published and unpublished studies. The analysis produced an asymmetric funnel plot, which indicated the presence of publication bias (**Fig 7**). Moreover, Egger's test revealed a significant publication bias (P < 0.001)(**SF 3 Table**).

## Nonparametric trim-and-fill analysis of the pooled prevalence of ESBL-producing Enterobacterales

A nonparametric trim-and-fill analysis was conducted to address the observed publication bias. After imputing data on the left, the pooled prevalence of ESBL-producing *Enterobacterales* among HIV-positive individuals across the globe remained stable at 20.30% (95% CI: 15.13–25.47%) (**SF 3 Table**). Conversely, when analyzing 21 studies with one data points imputed on the right, the pooled prevalence of ESBL-producing *Enterobacterales* among HIV-positive individuals was slightly higher at 21.58% (95% CI: 14.34–28.82%) (**SF 3 Table**).

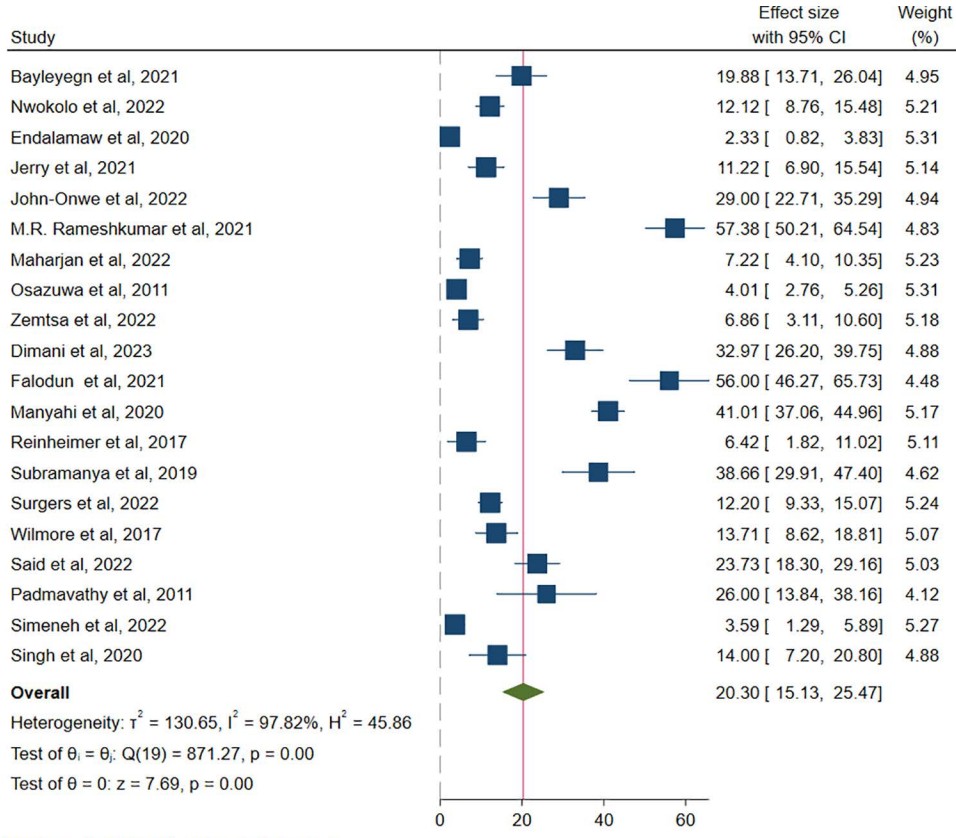

**Fig 2. Forest plot showing overall pooled prevalence of ESBL-producing *Enterobacterales* among HIV-positive individuals.**

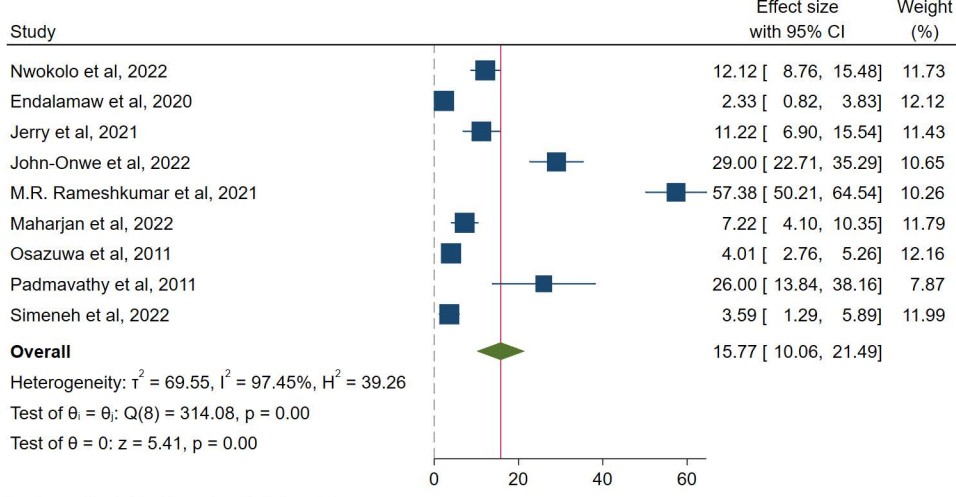

**Fig 3. Forest plot for the pooled prevalence of ESBL-producing *Enterobacterales* among infection case.**

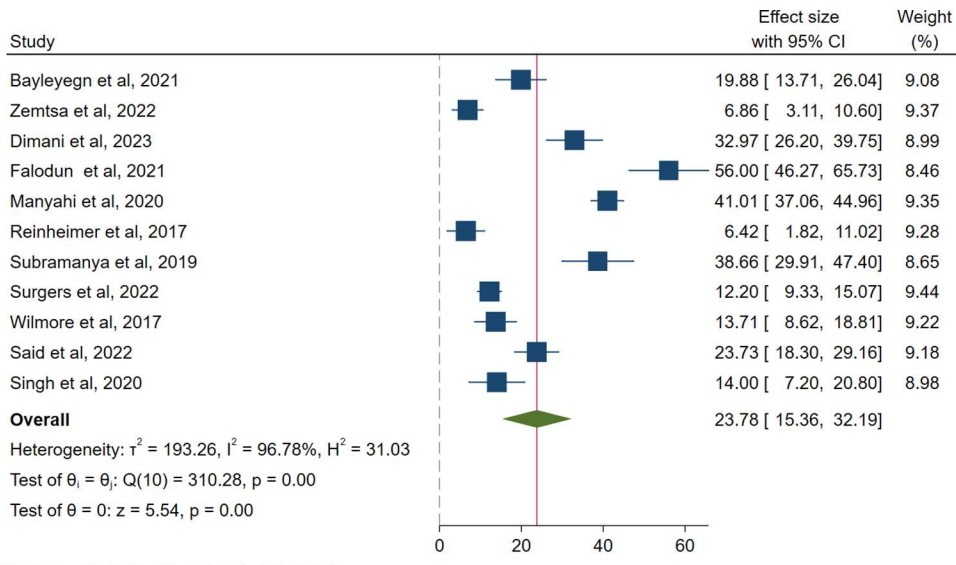

**Fig 4. Forest plot for the pooled prevalence of ESBL-producing *Enterobacterales* among colonization case.**

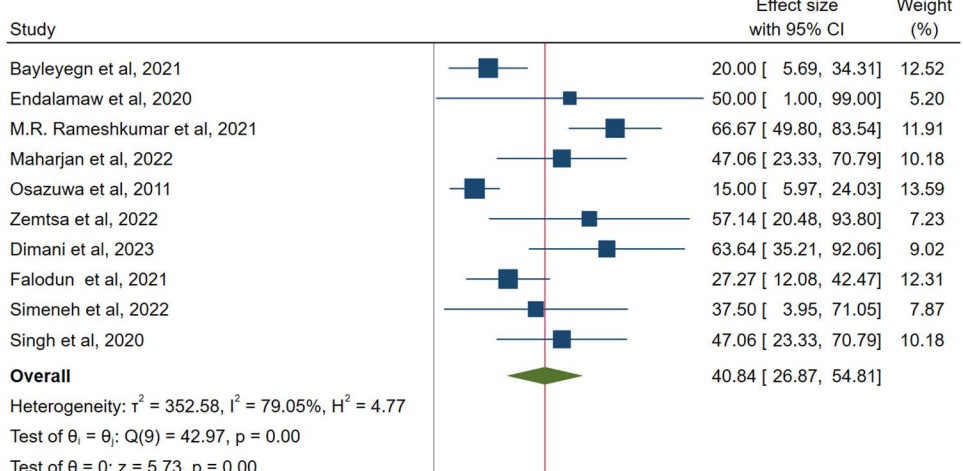

**Fig 5. Forest plot showing the pooled prevalence of ESBL-producing *K. pneumoniae* among HIV-positive individuals.**

## Sensitivity analysis

A sensitivity analysis was performed via a random effects model to assess the impact of individual studies on the combined estimate. The pooled prevalence of ESBL-producing pathogens that was obtained after individual studies were excluded was within the 95% CI of the total combined estimate. This shows that no single study had an effect on the overall pooled prevalence (**SF 3 Fig** and Table).

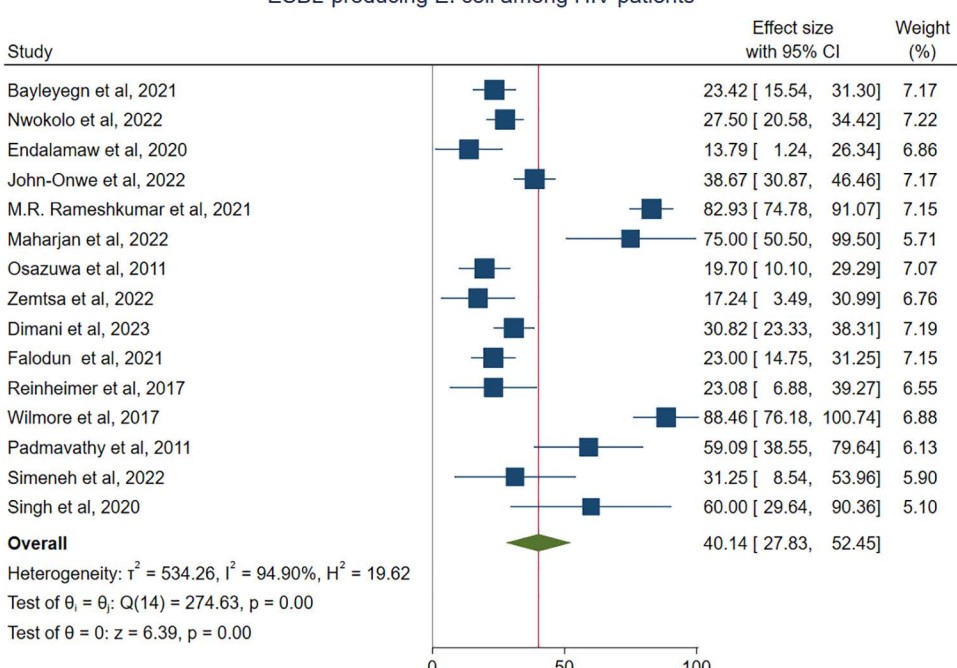

Fig 6. Forest plot showing the pooled prevalence of ESBL-producing *E. coli* among HIV-positive individuals.

## Subgroup analysis

To address heterogeneity, subgroup analysis was carried out by geographical location, publication year, age group, and phenotypic ESBL confirmation methods. When subgroup analysis was performed by continent, the highest pooled prevalence of ESBL-producing pathogens among HIV patients was observed in Asia, at 28.55% (195/715; 95% CI: 8.41–48.70, $I^2 = 97.85\%$, P<0.001), followed by Africa, at 19.12% (666/3981; 95% CI: 12.98–25.25, $I^2 = 98.07\%$, P<0.001). On the other hand, the lowest pooled prevalence was found in Europe, at 9.60% (68/609; 95% CI: 3.97–15.24, $I^2 = 77.07\%$, P=0.04) (**Fig 8**).

Similarly, subgroup analysis on the basis of the publication year of studies revealed that the highest pooled estimate was 21.53% (803/3904; 95% CI: 14.66–28.39, $I^2 = 98.12\%$, P<0.001) in and after 2020, whereas for studies published before 2020, the prevalence was 16.72% (128/1401; 95% CI: 7.17–26.27, $I^2 = 95.10\%$, P=0.001) (**Fig 9**).

In addition, a subgroup analysis was conducted based on patient age groups. The pooled prevalence of ESBL-producing *Enterobacterales* did not show a substantial difference across age categories. The prevalence was reported as 19.03% (416/2903; 95% CI: 12.99–25.07, $I^2 = 71.87\%$, P=0.03) in children, 19.57% (112/572; 95% CI: 8.39–30.75, $I^2 = 98.29\%$, P<0.001) in adults, and 21.25% (403/1830; 95% CI: 14.43–28.07, $I^2 = 97.88\%$, P<0.001) in studies involving all age groups (Fig 10).

Furthermore, a subgroup analysis was conducted based on the type of investigation method. The highest pooled prevalence of ESBL-producing pathogens among HIV-positive patients was reported in studies employing genotypic methods (24.33%, 657/2653; 95% CI: 15.02–33.65; $I^2 = 97.69\%$; P<0.001), followed by studies using phenotypic methods (15.42%, 274/2652; 95% CI: 10.53–20.31; $I^2 = 96.11\%$; P<0.001) (Fig 11).

We also performed subgroup analysis in terms of phenotypic ESBL detection methods. The highest pooled prevalence was observed using the double-disk synergy test (DDST), at 22.09% (569/3215; 95% CI: 14.80–29.37, $I^2 = 98.38\%$,

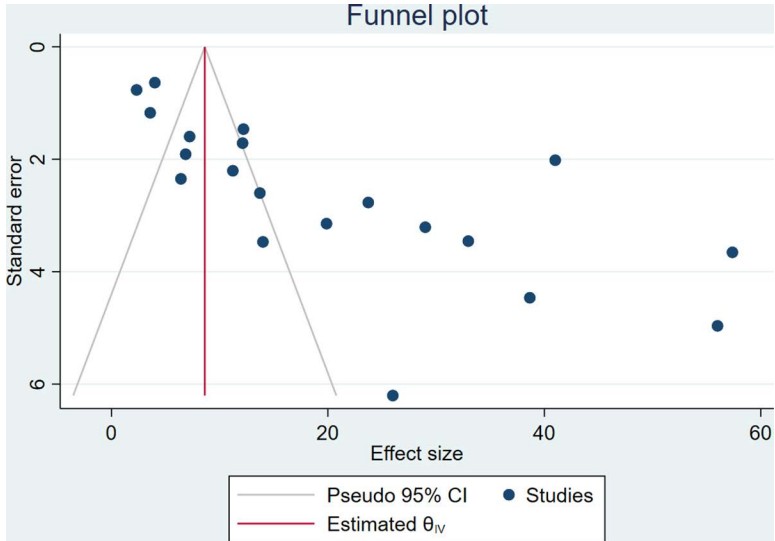

**Fig 7. Funnel plot showing publication bias for ESBL-producing *Enterobacterales* among HIV-positive individuals.**

P<0.001), followed by the combined disk test (CDT), at 19.70% (355/1981; 95% CI: 11.91–27.50, I²=96.04%, P<0.001). The lowest pooled prevalence was recorded using the Vitek-2 method, at 6.42% (7/109; 95% CI: 1.82–11.02, I²=0.0%, P<0.001) (**SF 5 Fig**).

### Risk factors for the prevalence of ESBL producers among HIV-positive individuals

A meta-analysis could not be performed because of the small number of studies reporting risk factors. Among the 20 included studies, only 4 reported risk factors; among these, variability in variables such as a history of antibiotic use, hospital admission, and low cluster of differentiation 4 (CD4) T lymphocyte counts was a risk factor for a higher prevalence of ESBL-producing Enterobacterales among HIV-positive individuals (Table 2).

### Types of ESBL enzyme-encoding genes via *Enterobacterales* among HIV-positive individuals

Although the included studies employed heterogeneous genotyping techniques, bla$_{CTX-M}$ (73/150, 48.7%) was the most prevalent genotype, followed by bla$_{TEM}$ (49/150, 32.7%) among the commonly reported ESBL enzyme-encoding genes, as documented in five studies (Table 3).

### Discussion

The management of antibiotic-resistant bacterial infections especially those caused by *Enterobacterales* poses a significant challenge in the context of infectious diseases like HIV [46]. Although not all ESBL-producing strains meet the strict definition of multidrug resistance (MDR), the production of ESBLs often confers resistance to multiple β-lactam antibiotics and is frequently associated with co-resistance to other antimicrobial classes. These pathogens are commonly linked to nosocomial infections [14]. In this context, our systematic review and meta-analysis aimed to estimate the global burden of ESBL-producing *Enterobacterales* among individuals living with HIV.

In our systematic review and meta-analysis, the overall pooled prevalence of ESBL-producing *Enterobacterales* among HIV-positive individuals was 20.30% (95% CI: 15.13–25.47%, P<0.001**),** with a high level of heterogeneity (I²=97.82%, p<0.001). This significant prevalence of ESBLs among these populations may be due to the increased number of antibiotics consumed and increased risk of acquisition of resistant pathogens during healthcare visits and hospitalizations [47].

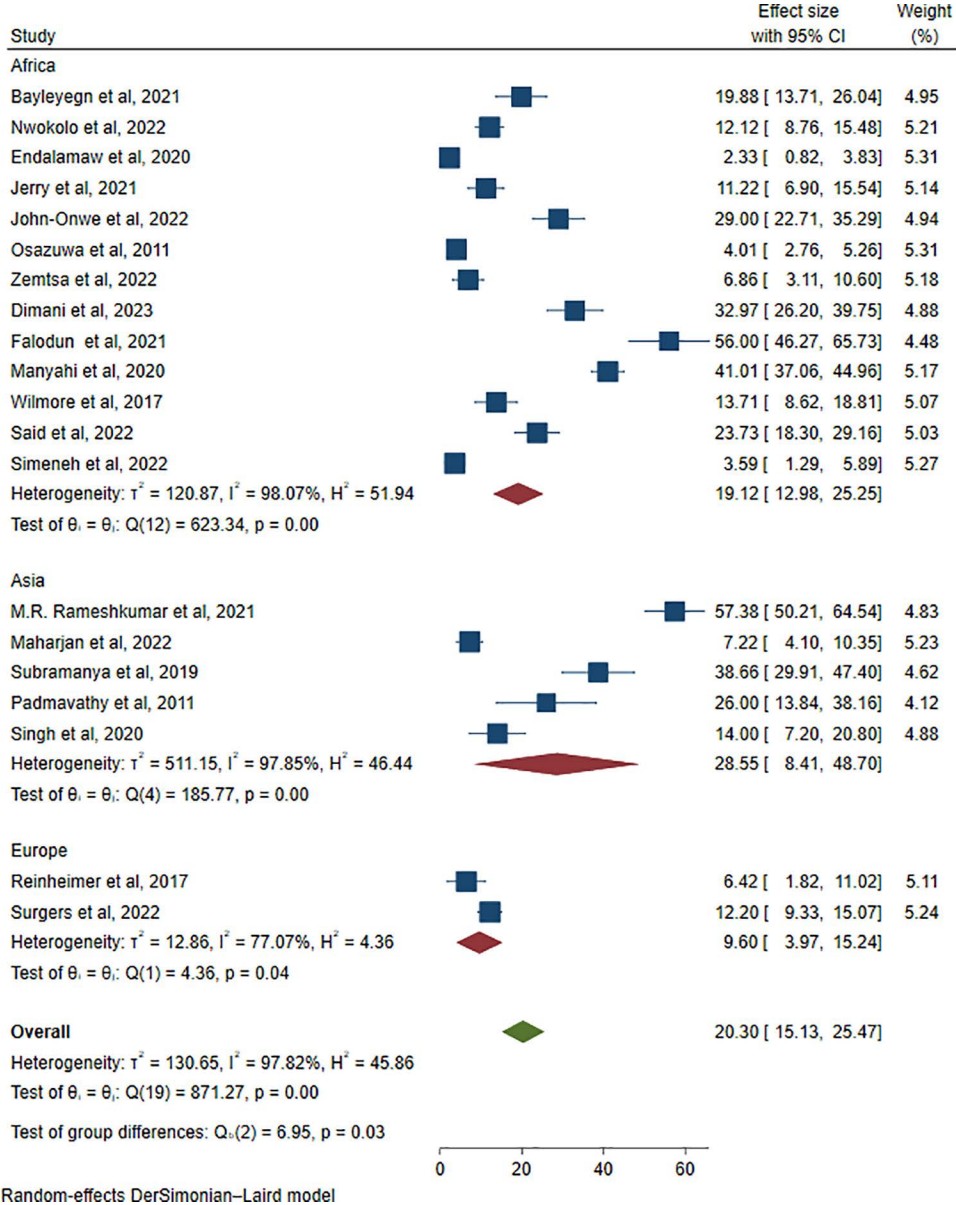

**Fig 8. Subgroup analysis of ESBL-producing *Enterobacterales* among HIV-positive individuals by continent.**

These populations are immunocompromised and need more frequent healthcare appointments and are vulnerable to comorbid conditions, which require hospital admissions, than are the general population [48].

The highest pooled prevalence of ESBL-producing pathogens among HIV-positive individuals was reported to be colonization 23.78% (95% CI: 15.36–32.19, I²=96.78%, p<0.001), followed by infection 15.77% (95% CI: 10.06–21.49, I²=97.45%, p<0.001). The higher prevalence may be explained by the methods used to report on colonization, sample size, method of detection, culture media. This may be because the gut microbiota is mostly composed of Enterobacterales, and HIV infection alters the composition of the microbiome and decreases the number of CD4+T cells in the gut-associated lymphoid tissue, which is associated with microbiota dysbiosis that favors colonization and subsequent infections with resistant strains [49,50].

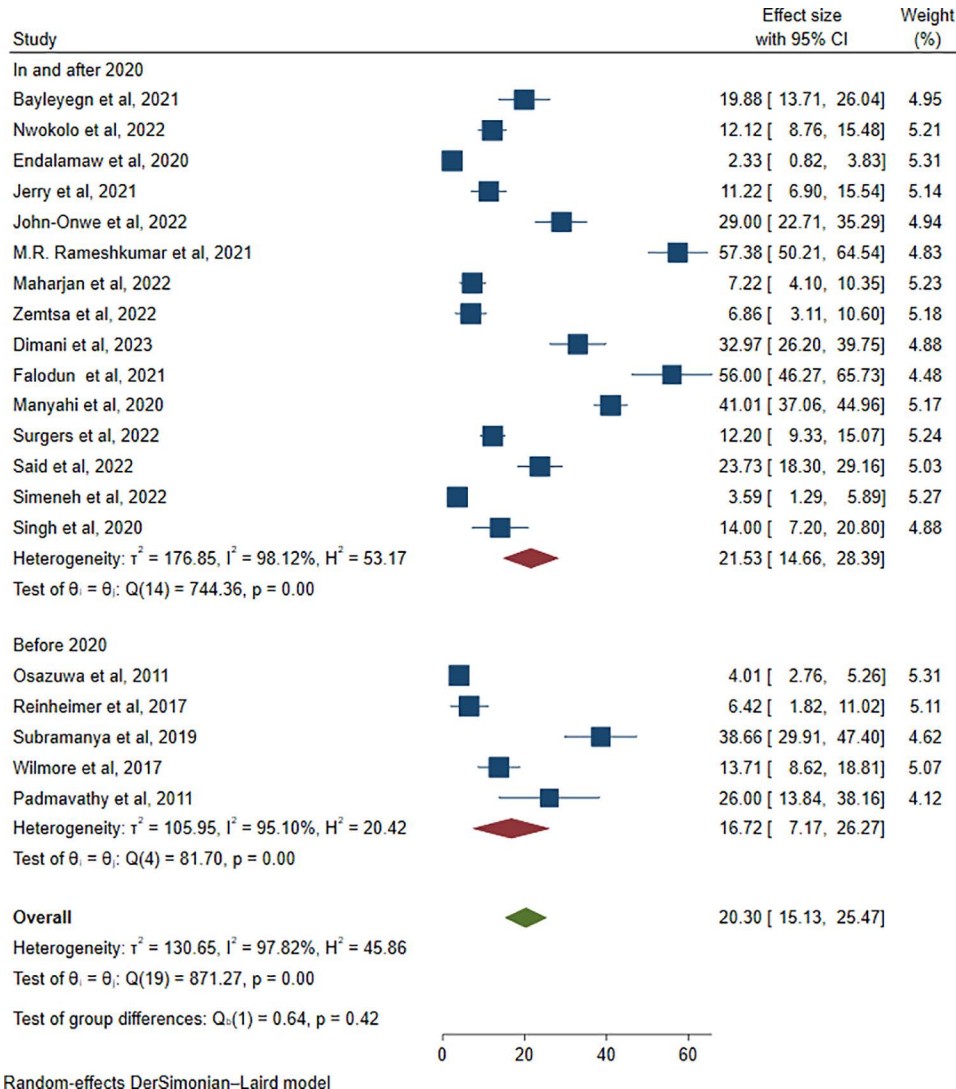

**Fig 9. Subgroup analysis of ESBL-producing *Enterobacterales* by year of publication.**

In the present meta-analysis, the predominant ESBL-producing pathogens among HIV-infected patients were *K. pneumoniae*, with a pooled prevalence of 40.84% (95% CI: 26.87–54.81%), followed closely by *E. coli* at 40.14% (95% CI: 27.83–52.45%). These organisms are known to resist beta-lactam antibiotics through the production of ESBL enzymes [51,52]. *K. pneumoniae* is most commonly associated with healthcare-associated infections, whereas, *E. coli* is typically linked to community-acquired infections. As a result, infections caused by these pathogens may contribute to increased complications in HIV-infected patients [53].

In this meta-analysis, a high level of heterogeneity ($I^2 = 97.82\%$) was observed. This is not surprising given the differences in study settings, patient statuses, age groups, sample types, publication years and methods used for the detection of ESBLs. To determine the possible sources of heterogeneity, subgroup analysis was performed by continent, and the highest pooled prevalence of ESBL-producing pathogens among HIV-positive individuals was reported in Asia: 28.55% (95% CI: 8.41–48.70, $I^2 = 97.85\%$, P < 0.001), followed by the African continent 19.12% (95% CI: 12.98–25.25, $I^2 = 98.07\%$, P < 0.001). However, the lowest pooled prevalence was found in Europe, at 9.60% (95% CI: 3.97–15.24, $I^2 = 77.07\%$,

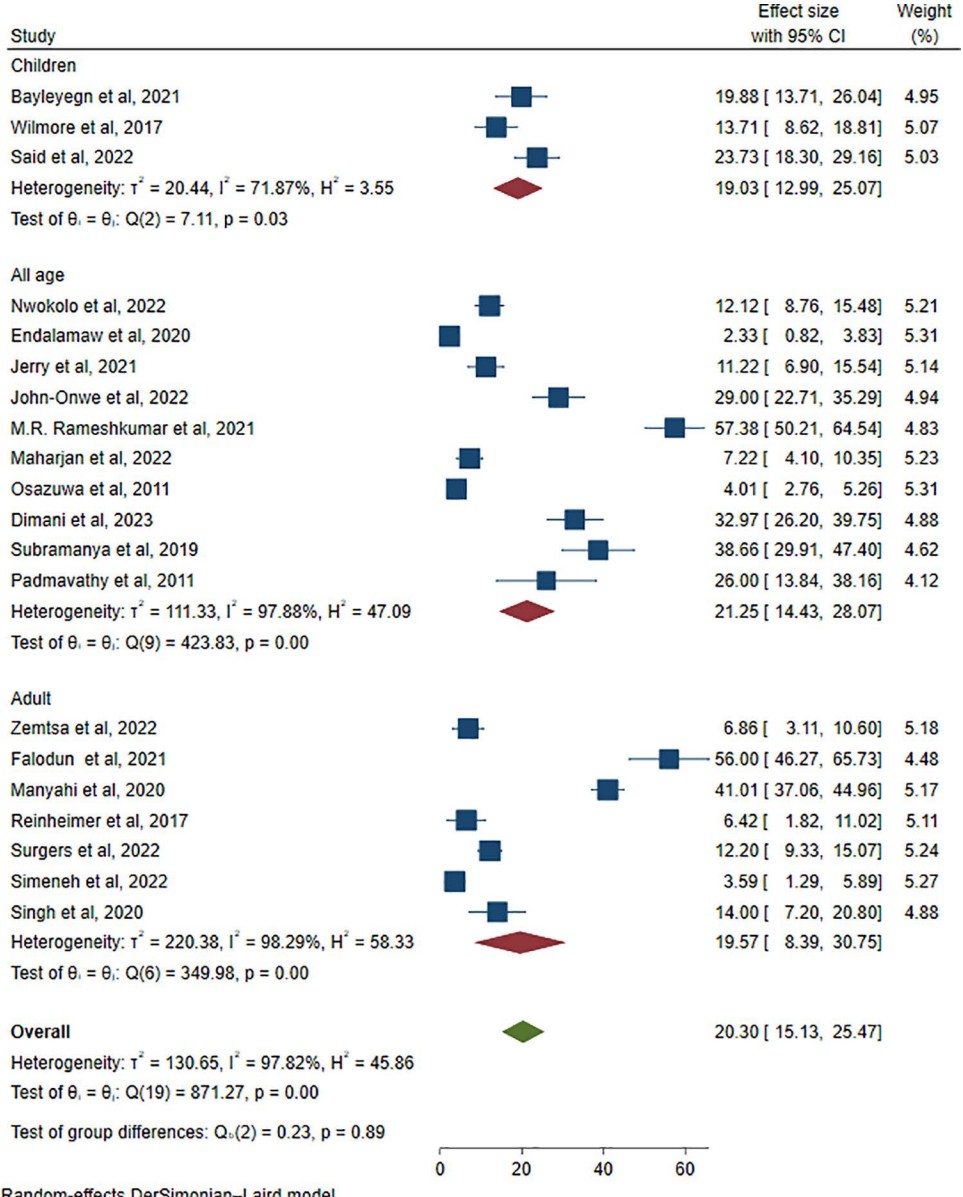

**Fig 10. Forest plot showing subgroup analysis of ESBL-producing *Enterobacterales* via age categories.**

P = 0.04). This is because the use of trimethoprim–sulfamethoxazole prophylaxis in developed nations is decreasing because of the early diagnosis of HIV and well-controlled antiretroviral therapy (ART), which reduces immunosuppression and bacterial infections [54].

However, the high burden of HIV infection and weak health systems to diagnose HIV early, inadequate adherence and poorly controlled ART make the immune system weakened and susceptible to bacterial infections in low-income countries. Additionally, as opportunistic infections are common among HIV-positive individuals in these countries, trimethoprim–sulfamethoxazole prophylaxis is widely used [55]. For this reason, this antibiotic can be used to coselect resistant strains among these populations [9]. Several studies have reported that trimethoprim-sulfamethoxazole prophylaxis is

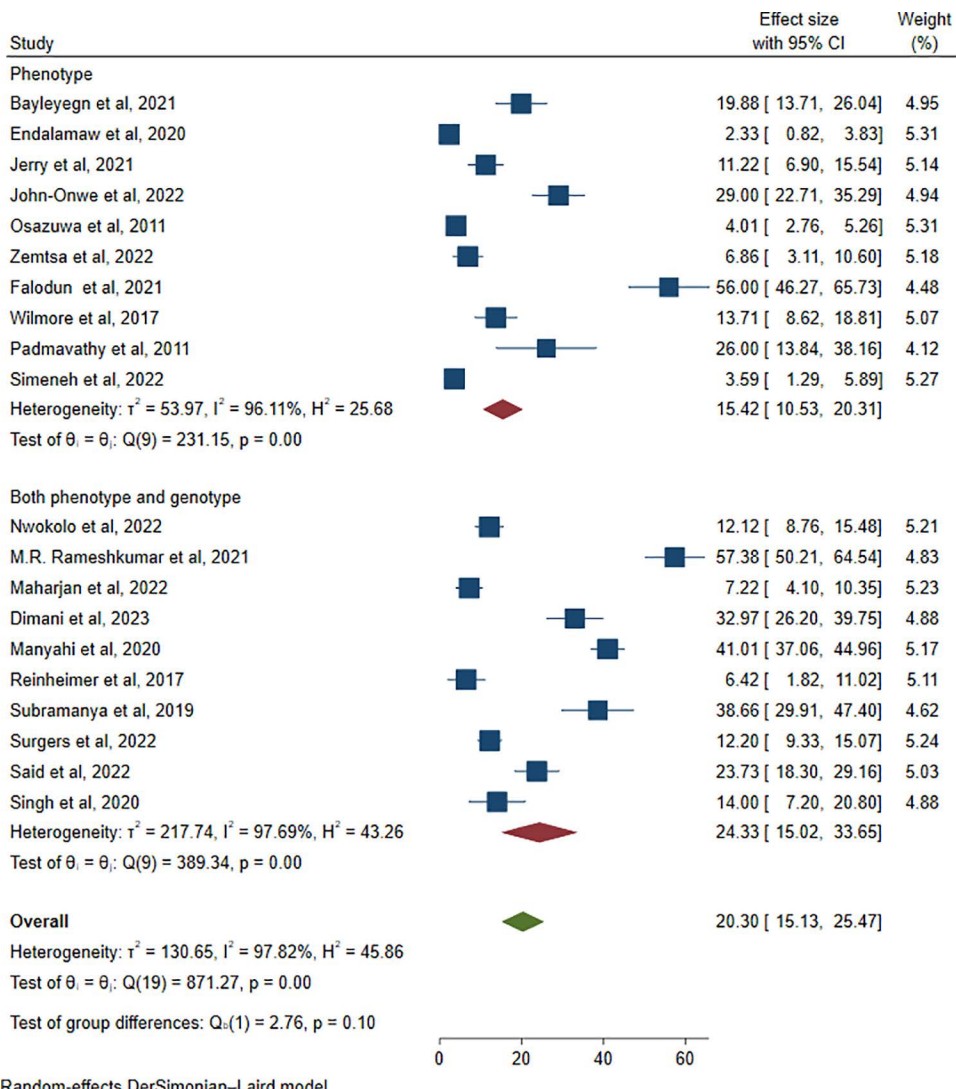

| Study | Effect size with 95% CI | Weight (%) |
|---|---|---|

**Phenotype**

| Study | Effect size with 95% CI | Weight (%) |
|---|---|---|
| Bayleyegn et al, 2021 | 19.88 [ 13.71, 26.04] | 4.95 |
| Endalamaw et al, 2020 | 2.33 [ 0.82, 3.83] | 5.31 |
| Jerry et al, 2021 | 11.22 [ 6.90, 15.54] | 5.14 |
| John-Onwe et al, 2022 | 29.00 [ 22.71, 35.29] | 4.94 |
| Osazuwa et al, 2011 | 4.01 [ 2.76, 5.26] | 5.31 |
| Zemtsa et al, 2022 | 6.86 [ 3.11, 10.60] | 5.18 |
| Falodun et al, 2021 | 56.00 [ 46.27, 65.73] | 4.48 |
| Wilmore et al, 2017 | 13.71 [ 8.62, 18.81] | 5.07 |
| Padmavathy et al, 2011 | 26.00 [ 13.84, 38.16] | 4.12 |
| Simeneh et al, 2022 | 3.59 [ 1.29, 5.89] | 5.27 |
| Heterogeneity: $\tau^2 = 53.97$, $I^2 = 96.11\%$, $H^2 = 25.68$ | 15.42 [ 10.53, 20.31] | |
| Test of $\theta_i = \theta_j$: Q(9) = 231.15, p = 0.00 | | |

**Both phenotype and genotype**

| Study | Effect size with 95% CI | Weight (%) |
|---|---|---|
| Nwokolo et al, 2022 | 12.12 [ 8.76, 15.48] | 5.21 |
| M.R. Rameshkumar et al, 2021 | 57.38 [ 50.21, 64.54] | 4.83 |
| Maharjan et al, 2022 | 7.22 [ 4.10, 10.35] | 5.23 |
| Dimani et al, 2023 | 32.97 [ 26.20, 39.75] | 4.88 |
| Manyahi et al, 2020 | 41.01 [ 37.06, 44.96] | 5.17 |
| Reinheimer et al, 2017 | 6.42 [ 1.82, 11.02] | 5.11 |
| Subramanya et al, 2019 | 38.66 [ 29.91, 47.40] | 4.62 |
| Surgers et al, 2022 | 12.20 [ 9.33, 15.07] | 5.24 |
| Said et al, 2022 | 23.73 [ 18.30, 29.16] | 5.03 |
| Singh et al, 2020 | 14.00 [ 7.20, 20.80] | 4.88 |
| Heterogeneity: $\tau^2 = 217.74$, $I^2 = 97.69\%$, $H^2 = 43.26$ | 24.33 [ 15.02, 33.65] | |
| Test of $\theta_i = \theta_j$: Q(9) = 389.34, p = 0.00 | | |

**Overall** — 20.30 [ 15.13, 25.47]

Heterogeneity: $\tau^2 = 130.65$, $I^2 = 97.82\%$, $H^2 = 45.86$

Test of $\theta_i = \theta_j$: Q(19) = 871.27, p = 0.00

Test of group differences: $Q_b(1) = 2.76$, p = 0.10

Random-effects DerSimonian–Laird model

**Fig 11. Forest plot showing subgroup analysis of ESBL-producing *Enterobacterales* via investigation methods.**

mostly associated with increased non-susceptibility of *Enterobacterales* to beta-lactam antibiotics and an increased risk of ESBL-producing *Enterobacterales* [9,56,57].

Another suggested reason for the highest prevalence of ESBL-producing pathogens among HIV-positive individuals in Asia and Africa is the poor antibiotic stewardship program and low control mechanism for antibiotic usage, which causes overuse and misuse of antibiotics in healthcare facilities that facilitate resistant infections [58]. Additionally, poor infection prevention mechanisms, such as poor hygiene and sanitation practices, environmental contamination and inadequate decontamination of medical devices, aggravate colonization and infection with resistant pathogens among HIV-positive individuals in developing countries [58,59].

According to the subgroup analysis by publication year, the highest pooled estimate of 21.53% (95% CI: 14.66–28.39, $I^2 = 98.12\%$, P < 0.001) was observed in studies published in and after 2020, whereas before 2020, the prevalence was 16.72% (95% CI: 7.17–26.27, $I^2 = 95.10\%$, P = 0.001). This gap may be due to the emergence of multidrug-resistant

**Table 2. Significant risk factors in some of the included studies.**

| Study | Risk factors assessed | Analysis | Significant risk factors | Odds ratio (95% CI) |
|---|---|---|---|---|
| Subramanya et al, 2019 [30] | Age, sex, occupation, CD4 count, ART, ART duration, admitted to hospital in at 6 m, history of antibiotic use in at 6 m, animal contact (domestic and livestock), using medication over-the-counter, international travel, food habit | Multivariate | With ART | 1.537 (0.349,6.762) |
| | | | Admitted to hospital in the previous 6 m | 3.638 (0.793,16.68) |
| | | | Close contact with livestock | 1.153 (0.515,2.581) |
| | | | Using medication over-the-counter | 1.988 (0.898,4.401) |
| Bayleyegn et al, 2021 [39] | Age, sex, residence, educational status, opportunistic infections, presence of fever, WHO stage of HIV, ART type, ART duration, history of antibiotic use, viral load, presence of diarrhea, eating uncooked products, eating row vegetable | Multivariate | History of antibiotic use | 3.2 (1.05–9.9) |
| Manyahi et al, 2020 [23] | Age, sex, residence, educational status, WHO stage of HIV, CD4 count, admitted to hospital in at 12 m, history of antibiotic use in at 1 m | Multivariate | CD4 count <350 | 1.78 1.03–3.09 |
| | | | History of antibiotic use in the previous 1 m | 1.55 1.08–2.22 |
| Wilmore et al, 2017 [29] | Age, gender, CD4, viral load, ART duration, admitted to hospital with pneumonia in last 12 m, admitted to hospital in at 12 m | Multivariate | With ART ≤ 1year | 8.47 (2.22–2.27) |
| | | | Admitted to hospital with pneumonia in last 12 m | 8.47 (1.12–64.07) |

Note: ART = antiretroviral therapy, HIV = human immunodeficiency virus, WHO = World Health Organization, CL = confidence interval, M = month, CD4 = cluster of differentiation 4.

**Table 3. Types of genes in some studies that carry out genotypic analysis.**

| Study (Author, Year) | No of isolates subjected to genotype test | Types of isolates | Total ESBL gene | blaSHV | blaTEM | blaCTX-M | blaCTX-M+blaTEM |
|---|---|---|---|---|---|---|---|
| Nwokolo et al, 2022 [27] | 44 | E. coli | 26 | 7 (26.9) | 8 (30.8) | 11 (42.3) | |
| Maharjan et al, 2022 [26] | 12 | E. coli | 9 | – | 2 (22.2) | 3 (33.3) | 4 (44.5) |
| | 17 | K.pneumoniae | 7 | – | 0 | 3 (42.9) | 4 (57.1) |
| 12 et al, 2022 [45] | 14 | E. coli | 12 | – | 6 (50.0) | 1 (8.3) | 5 (41.7) |
| | 5 | K. pneumoniae | 4 | – | – | 4 (100) | – |
| Dimani et al, 2023 [20] | 45 | E. coli | 63 | 3 (4.7) | 27 (42.9) | 33 (52.4) | – |
| | 11 | K. pneumoniae | 12 | 4 (33.3) | 3 (25.0) | 5 (41.7) | – |
| Singh et al, 2020 [28] | 10 | E. coli | 9 | 0 | 2 (22.2) | 7 (77.8) | – |
| | 17 | K. pneumoniae | 8 | 1 (12.5) | 1 (12.5) | 6 (75.0) | – |
| **Total** | 175 | – | 150 | 15 (10.0) | 49 (32.7) | 73 (48.7) | 13 (8.6) |

Note: ESBL = extended-spectrum β-lactamase, bla$_{TEM}$ = Temoneira β-lactamase, bla$_{SHV}$ = sulfhydryl reagent variable β-lactamase, bla$_{CTX-M}$ = cefotaxim-hydrolizing β-lactamase..

bacteria because the prevalence of ESBL-producing bacteria has increased over time, and recently studied papers may include a high number of ESBL-producing pathogens [60].

Furthermore, the subgroup analysis was carried out based on the investigation methods. Accordingly, the highest pooled prevalence of ESBL-producing pathogens among HIV patients was reported in studies that included genotypic investigations (24.33%, 95% CI: 15.02–33.65, I²=97.69%, P<0.001), followed by phenotypic investigations (15.42%, 95% CI: 10.53–20.31, I²=96.11%, P<0.001). The higher pooled prevalence of ESBL-producing pathogens reported in studies using genotypic methods may be attributed to the greater sensitivity and specificity of these techniques in detecting resistance genes, including those that are silent or weakly expressed. In contrast, phenotypic methods may underestimate prevalence due to their reliance on observable resistance, which can be affected by gene expression levels or overlapping

resistance mechanisms. Additionally, genotypic studies may have been conducted in settings with higher antibiotic pressure or more advanced diagnostic capacity, contributing to the observed differences [61].

In our systematic review, among the 20 included studies, only 2 reported that a history of antibiotic use in the previous month was a risk factor for infection and/or colonization with ESBL-producing pathogens among HIV-positive individuals. Additionally, the highest prevalence of ESBL-producing pathogens was reported in HIV-positive individuals who were admitted to the hospital for the last 6 and 12 months. Several previous studies identified prior hospitalization and antibiotic exposure as risk factors for antibiotic-resistant infections [12,47,62]. Patients may acquire resistant pathogens from hospitals, which may be due to selective pressure; when patients use antibiotics, sensitive bacterial strains are eliminated, leaving behind or selecting those variants that can resist them [47]. As cephalosporins are the most commonly used antibiotics for the treatment of various infections, they favor ESBL-producing strains [63].

Additionally, some studies have shown that taking ART for less than one year is a risk factor for ESBL incidence, and the risk of infection with ESBL-producing pathogens among HIV-positive individuals varies according to the CD4 count and is greatest in HIV-positive individuals with CD4 counts <350 cells/mm$^3$. Low immunity with low CD4+T cells is associated with an increased risk of opportunistic infections, increasing the likelihood of antimicrobial use and hospitalization. Consequently, this phenomenon predisposes resistant pathogens to the emergence of ESBL-producing strains [64].Among the various ESBL enzyme-encoding genes, bla$_{CTX-M}$ is the most prevalent (48.7%), followed by bla$_{TEM}$ (32.7%). The dominance of bla$_{CTX-M}$ genes is largely attributed to their efficient dissemination through mobile genetic elements such as plasmids, transposons, insertion sequences, and integrons, which facilitate the horizontal transfer of resistance genes among *Enterobacterales* and other gram-negative bacilli in clinical settings worldwide [65]. In developing countries, where infection prevention and control measures are often limited, this gene transfer is even more widespread occurring not only in humans but also in animals and the environment [66]. Because these mobile genetic elements frequently carry multiple resistance determinants, pathogens harboring ESBL genes are also more likely to acquire additional MDR traits [65,66]. As a result, infections caused by these organisms are harder to treat. Therefore, infection with these pathogens complicates treatment options and intensifies the level of care required for immunocompromised patients, such as those living with HIV.

## Limitations

Study restriction by language and year of publication are the main limitations of this study. Furthermore, few countries are overrepresented, as the majority of studies are from Africa, specifically, sub-Saharan Africa; however, owing to the high burden of HIV in this region, accurate generalization is difficult. Since a few studies reported the risk factors and the heterogeneous variables used, a meta-analysis was not performed on the associated factors.

## Conclusion and recommendations

This study revealed a significant prevalence of ESBL-producing *Enterobacterales* among HIV-positive individuals, with *K. pneumonia* and *E. coli* being the dominant ESBL producers. The highest magnitude of ESBL-producing pathogens was observed in Asia and Africa. With respect to the types of ESBL enzyme-encoding genes, the most prevalent was bla$_{CTX-M}$ followed by bla$_{TEM}$. Equitable allocation of resources across all regions is needed to reduce mortality due to complications in HIV patients with severe bacterial infections; the regions with the highest prevalence rates should pay attention to early diagnosis of HIV, adequate adherence and well-controlled ART. Consequently, this helps to reduce immunosuppression and bacterial infections. Early identification of infections and colonization with antibiotic-resistant pathogens among HIV patients is also needed.

## Supporting information files

**SF 1. Table, PRISMA checklist.**
(DOCX)

**SF 2. Table, quality assessment of the studies included in a systematic review and meta-analysis.**
(DOCX)

**SF 3. Doc, Publication bias and Sensitivity analysis.**
(DOCX)

**SF 4. Figs, forest plots showed the pooled prevalence of ESBL by each bacterial species.**
(DOCX)

**SF 5. Doc, Subgroup analysis.**
(DOCX)

**SF 6. Table for all studies identified in the literature search.**
(DOCX)

## Author contributions

**Conceptualization:** Mitkie Tigabie.

**Data curation:** Mitkie Tigabie, Abebe Birhanu, Getu Girmay, Kebebe Tadesse.

**Formal analysis:** Mitkie Tigabie.

**Funding acquisition:** Mitkie Tigabie.

**Investigation:** Mitkie Tigabie.

**Methodology:** Mitkie Tigabie, Abebe Birhanu, Muluneh Assefa, Kebebe Tadesse.

**Project administration:** Mitkie Tigabie.

**Resources:** Mitkie Tigabie.

**Software:** Mitkie Tigabie.

**Supervision:** Abebe Birhanu, Muluneh Assefa, Getu Girmay, Kebebe Tadesse.

**Validation:** Mitkie Tigabie, Kebebe Tadesse.

**Visualization:** Abebe Birhanu, Muluneh Assefa, Getu Girmay, Kebebe Tadesse.

**Writing – original draft:** Mitkie Tigabie.

**Writing – review & editing:** Mitkie Tigabie, Abebe Birhanu, Muluneh Assefa, Getu Girmay, Kebebe Tadesse.

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
