## [Decision Letter · Decision Letter 0]

30 Mar 2025

Dear Dr. Tigabie,

Thank you for submitting your manuscript to PLOS ONE. After careful consideration, we feel that it has merit but does not fully meet PLOS ONE’s publication criteria as it currently stands. Therefore, we invite you to submit a revised version of the manuscript that addresses the points raised during the review process.

We look forward to receiving your revised manuscript.

Kind regards,

Amir Nutman

Academic Editor

PLOS ONE

Journal Requirements:

2. We note that your Data Availability Statement is currently as follows: All relevant data are within the manuscript and in Supporting Information files.

5. Please include captions for your Supporting Information files at the end of your manuscript, and update any in-text citations to match accordingly. Please see our Supporting Information guidelines for more information: http://journals.plos.org/plosone/s/supporting-information .

6. As required by our policy on Data Availability, please ensure your manuscript or supplementary information includes the following:

Additional Editor Comments:

Dear Authors,

Thank you for submitting your manuscript, “Extended-spectrum β-lactamase-producing gram-negative bacilli among people living with human immunodeficiency virus across the globe: a systematic review and meta-analysis” to PLOS ONE. Based on the reviewers’ comments, I invite you to submit a revised version.

Please address all reviewer points in a point-by-point response and highlight changes in the manuscript. This decision does not guarantee acceptance, but I believe the work has potential pending revision.

I look forward to your resubmission.

Best regards,

Amir

Reviewers' comments:

Reviewer's Responses to Questions

**Comments to the Author**

1. Is the manuscript technically sound, and do the data support the conclusions?

Reviewer #1: No

Reviewer #2: Partly

2. Has the statistical analysis been performed appropriately and rigorously?

Reviewer #1: Yes

Reviewer #2: Yes

3. Have the authors made all data underlying the findings in their manuscript fully available?

Reviewer #1: Yes

Reviewer #2: Yes

4. Is the manuscript presented in an intelligible fashion and written in standard English?

Reviewer #1: Yes

Reviewer #2: Yes

Reviewer #1: The authors conducted a systematic review and meta-analysis on the prevalence of ESBL in Gram negative bacteria among people with HIV. While the methods used for the analysis are appropriate, I have several concerns with regards to how the data are interpreted and reported.

Firstly, the main results on the prevalence of ESBL should be reported according to bacterial species and type (colonisation vs. infection). It is known that there are differences in ESBL prevalence between common Gram negatives such as E. coli and K. pneumoniae and pooling results across organisms is less meaningful. Further the authors should present as a main analysis (instead of a subgroup) the prevalence of ESBL in colonisation and infection samples.

Secondly the authors should note that for A. baumanii and P. aeruginosa, there are other mechanisms that play a more important role in beta-lactam resistance than ESBL-production. For this reason, the authors should restrict the focus of the manuscript on Enterobacterales and exclude A baumanii and P. aeruginosa.

Lastly, the authors should revise the manuscript to improve clarity and avoid repetitions. More specific comments are added below.

Lines 62-69: use concordant referencing. Currently you are reporting on data from various sources and the figures are somewhat conflicting. The O’Neill report which is almost 10 years old may have overestimated the number of deaths due to AMR. The paragraph would be easier to read if only one set of figures per point made would be used. Also in line 66 the sentence moves from economic losses in the US to global economic losses which is confusing.

Line 72: “Furthermore, the surge of infections that cause immune suppression, such as human immunodeficiency virus “ consider rephrasing as the infection you are referring to is HIV and there is no surge of multiple infections associated with immunosuppression. (unless you mean conditions associated with immunosuppression e.g. long-term steroids etc.

Lines 82-84: introduce the epidemiology of HIV before introducing how it relates to resistance development

Line 90: “Among bacterial pathogens, gram negative bacilli are the most dangerous …” while it is true that gram-negative organisms pose more problems because of their resistance to antibiotics, they are not necessarily more dangerous (unless a reference is provided to support that statement e.g. that bacterial infections due to gram-negatives cause more deaths).

Line 95: “cephalosporins, and aztreonam, which are easily available” remove aztreonam from the listing as it is not easily available and commonly used

Line 97: “Enterobacteriaceae, Pseudomonas aeruginosa (P. aeruginosa), and Acinetobacter baumannii (A. baumannii) are the predominant gram-negative bacilli that produce ESBL enzymes “ please revise the sentence – the main mechanisms associated with resistance for P aeruginosa and A. baumanii are not ESBL-production.

Lines 105-107 – “the number of ESBL-producing gram-negative bacilli among HIV-positive individuals varies from 2.3% to 65.6%” – these are proportions not numbers

Lines 109-110: “On the one hand, various studies in the past reported higher ESBLs among HIV-positive individuals [20-29]. On the other hand, some studies reported that ESBL producers were more common among HIV-negative patients than among HIV-positive individuals [30-32].” – please consolidate these sentences to avoid repetitions

Line 139: please clarify in the methods whether the review included studies reporting exclusively on people living with HIV or it also included studies with mixed populations which reported on AMR prevalence by HIV status

Line 157: “The three reviewers listed above, MT, KT, and AB, were independently assessed to assess the quality of the included studies.” Please revise to avoid repetition

Figure 1: conventionally in the PRISMA diagram, the studies are reported separately as to how many were excluded in the title and abstract screening (reason for exclusion not necessary) and how many were excluded in the full text phase (with reason for exclusion).

Line 214 and above: be specific in the methods on how the prevalence was calculated for case-control studies (depending on the study design, might only be possible to calculate prevalence in the cases)

Table 1: please check and/or comment on including studies on infection which reported on stool samples. In this setting ESBL-producing organisms are also likely to represent colonisation (e.g. the study by Falodun reference #23 et al is clearly reporting on colonisation). Also check as there is an error with reference #22 which is a different study by Falodun et al. (not Israel)

Line 231: please note that it is not appropriate to pool prevalence across very different bacterial species and sample types. At least the pooling should be done separately for colonisation and infection and ideally separate for the main Enterobacterales species. This should not be treated as a subgroup analysis but rather the main analysis. I would suggest that you omit Pseudomonas and Acinetobacter completely from the analysis and manuscript as these organisms have additional mechanisms causing beta-lactam resistance, other than ESBL.

Table 2: “Rate of ESBL…” heading – please rename as this is not a rate (prevalence).

Table 2: Comment separately in the manuscript text on the prevalence of ESBL in Salmonella – 20% is quite high and concerning for clinical practice.

Table 3: can be omitted

Figure 3 should be redrawn as it is difficult to read with a black background. Also the numbers on the x-axis are overlapping. The figure can be included in the supplement.

Line 279: the subgroup analysis on method used can be omitted (or included in the supplement). For the other subgroup analyses please see my comment above. The authors could also add a subgroup analysis by organism and type (colonisation vs. infection) for community-acquired and hospital associated infections

Table 5 and Supplement: the genes should be reported according to the bacterial species in which they were identified (consider reporting for E. coli and K. pneumoniae only). Please also specify which SHV/CTX-M/TEM/OXA genes were reported as not all of them are ESBL

Line 322: please rephrase. The organisms are not multidrug resistant because they are ESBL-producers (there is a somewhat old definition of MDR and being ESBL does not suffice).

Line 339: please revise as E. coli is a common cause of community-acquired infections

Line 377: the higher prevalence may be explained by the methods used to report on colonisation (denominators, method of detection – culture media).

Lines 401-410: paragraph has repetitive sentences please revise

Reviewer #2: This systematic review and meta-analysis provide valuable insights into ESBL prevalence among HIV-positive individuals. While the study is methodologically strong, improvements in data selection, bias reduction, and clinical interpretation could enhance its impact. Future research should focus on treatment outcomes, stewardship interventions, and emerging resistance mechanisms to inform better infection control policies in HIV care settings.The study adheres to PRISMA guidelines, ensuring methodological transparency. It addresses a critical intersection of HIV and antimicrobial resistance (AMR), a growing concern for global health.

**Do you want your identity to be public for this peer review?** For information about this choice, including consent withdrawal, please see our Privacy Policy

Reviewer #1: No

Reviewer #2: **Yes: ** Rahul Garg

---

## [Author Response · Author response to Decision Letter 1]

17 Apr 2025

PONE-D-24-40800

Extended-spectrum β-lactamase-producing gram-negative bacilli among people living with human immunodeficiency virus across the globe: a systematic review and meta-analysis

PLOS ONE

Dear Dr. Tigabie,

Thank you for submitting your manuscript to PLOS ONE. After careful consideration, we feel that it has merit but does not fully meet PLOS ONE’s publication criteria as it currently stands. Therefore, we invite you to submit a revised version of the manuscript that addresses the points raised during the review process.

We look forward to receiving your revised manuscript.

Kind regards,

Amir Nutman

Academic Editor

PLOS ONE

Response to Academic Editor, Reviewers as well as Journal Requirements:

Authors: We appreciate for spending your precious time and forwarding your valuable comments, which have significantly improved our manuscript. We are also grateful for this positive feedback. Please see below, bold, for a point-by-point response to the reviewers. We've copied your comments and responses below to make things easier for you. All line numbers refer to the revised manuscript file.

Journal Requirements:

Authors: we have prepared the manuscript based on PLOS ONE's style requirements.

2. We note that your Data Availability Statement is currently as follows: All relevant data are within the manuscript and in Supporting Information files.

Authors: we have summited all in Supporting Information files.

Authors: we all authors decide and agreed on PLOS ONE's data sharing plan. All authors decide and agreed data availability statement.

Authors: we provided ethics statement in the Methods section of the manuscript.

Authors: we provided captions for the Supporting Information files at the end of the manuscript

6. As required by our policy on Data Availability, please ensure your manuscript or supplementary information includes the following:

Authors: we provided a table that included all studies identified in the literature search, along with reason(s) for exclusion for every excluded study. We uploaded this table in the Supporting Information files.

We also, provided a table showing the completed risk of bias as well as Name of data extractors and date of data extraction for included studies.

Additional Editor Comments:

Dear Authors,

Thank you for submitting your manuscript, “Extended-spectrum β-lactamase-producing gram-negative bacilli among people living with human immunodeficiency virus across the globe: a systematic review and meta-analysis” to PLOS ONE. Based on the reviewers’ comments, I invite you to submit a revised version.

Please address all reviewer points in a point-by-point response and highlight changes in the manuscript. This decision does not guarantee acceptance, but I believe the work has potential pending revision.

I look forward to your resubmission.

Best regards,

Amir

Response to Academic Editor

Authors: Thank you for your positive feedback; we appreciate your feedback. We have revised the entire manuscript as necessary and have attempted to address the comments from the reviewers.

Reviewers' comments:

Reviewer's Responses to Questions

Comments to the Author

1. Is the manuscript technically sound, and do the data support the conclusions?

Reviewer #1: No

Reviewer #2: Partly

2. Has the statistical analysis been performed appropriately and rigorously?

Reviewer #1: Yes

Reviewer #2: Yes

3. Have the authors made all data underlying the findings in their manuscript fully available?

Reviewer #1: Yes

Reviewer #2: Yes

4. Is the manuscript presented in an intelligible fashion and written in standard English?

Reviewer #1: Yes

Reviewer #2: Yes

5. Review Comments to the Author

Response to Reviewers

Authors: we would like to say thank you for reviewing our work and making insightful suggestions and comments that helped to strengthen our manuscript. We have revised the manuscript as necessary.

Reviewer #1: The authors conducted a systematic review and meta-analysis on the prevalence of ESBL in Gram negative bacteria among people with HIV. While the methods used for the analysis are appropriate, I have several concerns with regards to how the data are interpreted and reported.

Authors: Thank you for your positive feedback.

Reviewer #1: Firstly, the main results on the prevalence of ESBL should be reported according to bacterial species and type (colonisation vs. infection). It is known that there are differences in ESBL prevalence between common Gram negatives such as E. coli and K. pneumoniae and pooling results across organisms is less meaningful. Further the authors should present as a main analysis (instead of a subgroup) the prevalence of ESBL in colonisation and infection samples.

Authors: Thank you for bringing this issue to our intention. We have revised the analysis according to bacterial species and type (colonisation vs. infection). (Please refer to the revised manuscript's result section line # 248-272 and Supplementary File 4, Fig).

Reviewer #1: Secondly the authors should note that for A. baumanii and P. aeruginosa, there are other mechanisms that play a more important role in beta-lactam resistance than ESBL-production. For this reason, the authors should restrict the focus of the manuscript on Enterobacterales and exclude A baumanii and P. aeruginosa.

Authors: Thank you for your positive feedback. We have reanalyzed the data after excluding A baumanii and P. aeruginosa, now the manuscript focused on only about Enterobacterales (Please refer to the revised manuscript).

Reviewer #1: Lastly, the authors should revise the manuscript to improve clarity and avoid repetitions. More specific comments are added below.

Authors: Thank you for your input. We have accepted your comment and we tried to amend to improve clarity and avoid repetitions (Please refer to the revised manuscript).

Reviewer #1: Lines 62-69: use concordant referencing. Currently you are reporting on data from various sources and the figures are somewhat conflicting. The O’Neill report which is almost 10 years old may have overestimated the number of deaths due to AMR. The paragraph would be easier to read if only one set of figures per point made would be used. Also in line 66 the sentence moves from economic losses in the US to global economic losses which is confusing.

Authors: Thank you for raising this interesting point. We have changed the whole paragraph with updated data. (Please refer to the revised manuscript line # 65-72).

Reviewer #1: Line 72: “Furthermore, the surge of infections that cause immune suppression, such as human immunodeficiency virus “consider rephrasing as the infection you are referring to is HIV and there is no surge of multiple infections associated with immunosuppression. (unless you mean conditions associated with immunosuppression e.g. long-term steroids etc.

Authors: We appreciate your feedback. We have rephrased and corrected a typing error (Please refer to the revised manuscript line # 75-79).

Reviewer #1: Lines 82-84: introduce the epidemiology of HIV before introducing how it relates to resistance development

Authors: Thank you for raising this interesting point. We incorporated th

---

## [Decision Letter · Decision Letter 1]

27 Apr 2025

Dear Dr. Tigabie,

Thank you for submitting your manuscript to PLOS ONE. After careful consideration, we feel that it has merit but does not fully meet PLOS ONE’s publication criteria as it currently stands. Therefore, we invite you to submit a revised version of the manuscript that addresses the points raised during the review process.

We look forward to receiving your revised manuscript.

Kind regards,

Amir Nutman

Academic Editor

PLOS ONE

Journal Requirements:

**Additional Editor Comments:**

Thank you for your submission.

After reviewing the revised manuscript, I find it acceptable pending minor revisions:

1. The manuscript would benefit from language editing by a fluent English speaker to improve overall readability.

2. When reporting prevalence percentages, please also indicate the numerator and denominator (e.g., x/y, z%) to enhance clarity.

3. In Supplementary Table 2, only one author is listed as performing data extraction, which does not align with the description in lines 212–213 of the main text. Please correct this inconsistency.

Reviewers' comments:

Reviewer's Responses to Questions

**Comments to the Author**

Reviewer #1: All comments have been addressed

2. Is the manuscript technically sound, and do the data support the conclusions?

Reviewer #1: Yes

3. Has the statistical analysis been performed appropriately and rigorously?

Reviewer #1: Yes

4. Have the authors made all data underlying the findings in their manuscript fully available?

Reviewer #1: Yes

5. Is the manuscript presented in an intelligible fashion and written in standard English?

Reviewer #1: Yes

Reviewer #1: All my previous comments have been addressed.

However I am slightly confused by the different prevalences reported - perhaps because of numerators and denominators "The predominant ESBL producers were K. pneumoniae, with a pooled prevalence of 40.84% (95% CI: 26.87–54.81%), followed closely by E. coli at 40.14% (95% CI: 27.83–52.45%). In the subgroup analysis, the highest magnitude of ESBL producing pathogens was observed in Asia (28.5534.97%), followed by Africa (19.1220.75%). Additionally, the highest pooled prevalence of ESBL-producing pathogens among HIV-positive individuals was reported to be colonization 23.78% (95% CI: 15.36–32.19, I² = 96.78%, p <0.001), followed by infection 15.77% (95% CI: 10.06–21.49, I² = 97.45%, p < 0.001). "

The pooled prevalence of ESBL E coli is reported above at 40% and below at 15-23%. Also for the subgroup analyses by country there are prevalences around 20% by continent while overall it is 40%.

**Do you want your identity to be public for this peer review?** For information about this choice, including consent withdrawal, please see our Privacy Policy

Reviewer #1: No

---

## [Author Response · Author response to Decision Letter 2]

6 May 2025

Response to Academic Editor and Reviewers

Authors: We appreciate for spending your precious time and forwarding your valuable comments, which have significantly improved our manuscript. We are also grateful for this positive feedback. Please see below, bold, for a point-by-point response to Academic Editor and the reviewers. We've copied your comments and responses below to make things easier for you.

Academic Editor

1. The manuscript would benefit from language editing by a fluent English speaker to improve overall readability.

Authors: Thank you for your positive feedback. We have revised the entire manuscript with the assistance of a local editor and a professor to ensure appropriate language editing

2. When reporting prevalence percentages, please also indicate the numerator and denominator (e.g., x/y, z%) to enhance clarity.

Authors: Thank you for bringing this issue to our intention. We have provided the numerator and denominator of the prevalence result along with percentages to enhance clarity. Please do not be confused by the fact that the pooled prevalence is not identical to the crude proportion calculated. For example, the pooled prevalence of 20.30% is not identical to the crude proportion calculated as 931/5305. In meta-analysis, the pooled prevalence (20.30%) generated by STATA is not a simple average or direct proportion. Rather, it is weighted summary estimates derived from a meta-analysis model specifically, a random-effects model which accounts for variation across studies, differential weighting, and heterogeneity. Therefore, we kindly request that the prevalence results to be interpreted within this context.

3. In Supplementary Table 2, only one author is listed as performing data extraction, which does not align with the description in lines 212–213 of the main text. Please correct this inconsistency.

Authors: Thank you for bringing this to our attention. We have reviewed and corrected the inconsistency in the number of authors between Supplementary Table 2 and the main text.

Reviewer #1: All my previous comments have been addressed.

However I am slightly confused by the different prevalences reported - perhaps because of numerators and denominators "The predominant ESBL producers were K. pneumoniae, with a pooled prevalence of 40.84% (95% CI: 26.87–54.81%), followed closely by E. coli at 40.14% (95% CI: 27.83–52.45%). In the subgroup analysis, the highest magnitude of ESBL producing pathogens was observed in Asia (28.5534.97%), followed by Africa (19.1220.75%). Additionally, the highest pooled prevalence of ESBL-producing pathogens among HIV-positive individuals was reported to be colonization 23.78% (95% CI: 15.36–32.19, I² = 96.78%, p <0.001), followed by infection 15.77% (95% CI: 10.06–21.49, I² = 97.45%, p < 0.001). "

Authors: We appreciate your feedback. We have checked and updated prevalence results via provided the numerator and denominator of the prevalence result along with percentages to enhance clarity. However, Please do not be confused by the fact that the pooled prevalence is not identical to the crude proportion calculated. For example, the pooled prevalence of 20.30% is not identical to the crude proportion calculated as 931/5305. In meta-analysis, the pooled prevalence (20.30%) generated by STATA is not a simple average or direct proportion. Rather, it is weighted summary estimates derived from a meta-analysis model specifically, a random-effects model which accounts for variation across studies, differential weighting, and heterogeneity. Therefore, we kindly request that the prevalence results to be interpreted within this context.

Reviewer #1: The pooled prevalence of ESBL E coli is reported above at 40% and below at 15-23%. Also for the subgroup analyses by country there are prevalences around 20% by continent while overall it is 40%.

Authors: Thank you for reflecting on your concern. The overall pooled prevalence of ESBL-producing E. coli (e.g., 40%) represents a weighted average from all included studies, regardless of location, time, or population. Subgroup analyses by continent show averages within smaller groups, often with different numbers of studies, sample sizes, and local epidemiology. It's entirely possible for subgroups to show lower prevalence rates like 15–23% while the overall prevalence is higher, especially when High-prevalence studies (e.g., from specific countries) had large sample sizes or more weight in the analysis. Furthermore, there are fewer studies from low-prevalence areas, giving them less impact on the pooled estimate.

---

## [Decision Letter · Decision Letter 2]

19 May 2025

Extended-spectrum β-lactamase-producing Enterobacterales among people living with human immunodeficiency virus across the globe: a systematic review and meta-analysis

PONE-D-24-40800R2

Dear Dr. Tigabie,

We’re pleased to inform you that your manuscript has been judged scientifically suitable for publication and will be formally accepted for publication once it meets all outstanding technical requirements.

Kind regards,

Amir Nutman

Academic Editor

PLOS ONE

Additional Editor Comments (optional):

Reviewers' comments:

Reviewer's Responses to Questions

**Comments to the Author**

Reviewer #1: All comments have been addressed

2. Is the manuscript technically sound, and do the data support the conclusions?

Reviewer #1: Yes

3. Has the statistical analysis been performed appropriately and rigorously?

Reviewer #1: Yes

4. Have the authors made all data underlying the findings in their manuscript fully available?

Reviewer #1: Yes

5. Is the manuscript presented in an intelligible fashion and written in standard English?

Reviewer #1: Yes

Reviewer #1: The authors have provided responses to my comments (from the previous round of review) and I have no further comments.

**Do you want your identity to be public for this peer review?** For information about this choice, including consent withdrawal, please see our Privacy Policy

Reviewer #1: No

---

## [Editor Report · Acceptance letter]

PONE-D-24-40800R2

PLOS ONE

Dear Dr. Tigabie,

I'm pleased to inform you that your manuscript has been deemed suitable for publication in PLOS ONE. Congratulations! Your manuscript is now being handed over to our production team.

Kind regards,

on behalf of

Dr. Amir Nutman

Academic Editor

PLOS ONE